# Samples Are Useful? Not Always:
# denoising policy gradient updates using variance explained

## Abstract

Policy gradient algorithms in reinforcement learning optimize the policy directly and rely on efficiently sampling an environment. However, while most sampling procedures are based on sampling the agent's policy directly, self-performance measures could be used to improve sampling before each policy update. Following this line of thoughts, we propose SAUNA, a method where transitions are rejected from the gradient updates if they do not meet a particular criterion, and kept otherwise. The criterion, $\mathcal{V}^{ex}$, is the fraction of variance explained by the value function $V$: a measure of the discrepancy between $V$ and the returns. In this work, $\mathcal{V}^{ex}$ is used to estimate the impact transitions will have on learning: it refines sampling by simplifying the underlying state space and improves policy gradient methods. In this paper: (a) We introduce $\mathcal{V}^{ex}$, the criterion used for denoising policy gradient updates. We refer to this procedure as transition dropout and explore its implications on learning. (b) We conduct experiments across a variety of benchmark environments, including continuous control problems. SAUNA clearly improves performance. (c) We investigate how $\mathcal{V}^{ex}$ reliably selects samples with most positive impact on learning. (d) We study how this criterion can work as a dynamic tool to adjust the ratio between exploration and exploitation.

## 1 Introduction

Learning to control agents in simulated environments has been a challenge for decades in reinforcement learning (RL) (Nguyen & Widrow, 1990; Werbos, 1989; Schmidhuber & Huber, 1991; Robinson & Fallside, 1989) and has recently led to a lot of research efforts in this direction (Mnih et al., 2013; Burda et al., 2019; Ha & Schmidhuber, 2018; Silver et al., 2016; Espeholt et al., 2018), notably in policy gradient methods (Schulman et al., 2016; Silver et al., 2014; Lillicrap et al., 2016; Mnih et al., 2016). Despite the definite progress made, policy gradient algorithms still heavily suffer from sample inefficiency (Kakade, 2003; Wu et al., 2017; Schulman et al., 2017; Wang et al., 2017).

In particular, many policy gradient methods are subject to use as much experience as possible in the most efficient way. However, the quality of the sampling procedure also determines the final performance of the agent. We make the hypothesis that *not all experiences are worth* to use in the gradient update. Indeed, while perhaps trajectory simulations should be as rich as possible, some transitions may instead add noise to the gradient update, diluting relevant signals and hindering learning.

The central idea of SAUNA is to reject transitions that are not informative for the particular task at hand. For that purpose, we use a measure of the discrepancy between the estimated state value and the observed returns. This discrepancy is formalized with the notion of the fraction of variance explained $\mathcal{V}^{ex}$ (Kvålseth, 1985). Transitions for which $\mathcal{V}^{ex}$ is close to zero are those for which the correlation between the value function $V$ and the observed returns is also close to zero. SAUNA keeps transitions where there is either strong correlation or lack of fit between $V$ and the returns, while avoiding the learning signals to be diluted by the dropped out samples. We will examine the impact of transition dropout (inspired by Srivastava et al. (2014) and Freeman et al. (2019)) and its theoretical implications. We consider on-policy methods for their unbiasedness and stability

compared to off-policy methods (Nachum et al., 2017). However, our method can be applied to off-policy methods as well, and we leave this investigation open for future work.

The contributions of this paper are summarized as follows:

1. We propose to move from a simple policy-based sampling procedure based to one taking into account the agent's ability in an environment measured by $\mathcal{V}^{ex}$. We explore how the use of $\mathcal{V}^{ex}$ can drive the alignment between the samples used to update the policy and the agent's progress, while simplifying the underlying state space with transition dropout.

2. We provide a method that transforms policy gradient algorithms by assuming that not all samples are useful for learning and that these disturbing samples should, therefore, be rejected. While our method is a simple extension of policy gradient algorithms, it adds a variance criterion to the optimization problem and introduces a novel rejection sampling procedure.

3. By combining (1) and (2), we obtain a learning algorithm that is empirically effective in learning neural network policies for challenging control tasks. Our results extend the state-of-the-art in using reinforcement learning for high-dimensional continuous control. We also evaluate the theoretical implications of our method.

## 2    PRELIMINARIES

We consider a Markov Decision Process (MDP) with states $s \in \mathcal{S}$, actions $a \in \mathcal{A}$, transition distribution $s_{t+1} \sim P(s_t, a_t)$ and reward function $r(s, a)$. Let $\pi = \{\pi(a|s), s \in \mathcal{S}, a \in \mathcal{A}\}$ denote a stochastic policy and let the objective function be the traditional expected discounted reward:

$$J(\pi) \triangleq \mathbb{E}_{\tau \sim \pi} \left[ \sum_{t=0}^{\infty} \gamma^t r\left(s_t, a_t\right) \right],  \tag{1}$$

where $\gamma \in [0, 1)$ is a discount factor (Puterman, 1994) and $\tau = (s_0, a_0, s_1, \dots)$ is a trajectory sampled from the environment. Policy gradient methods aim at modelling and optimizing the policy directly (Williams, 1992). The policy $\pi$ is generally implemented with a function parameterized by $\theta$. In the sequel, we will use $\theta$ to denote the parameters as well as the policy. In deep reinforcement learning, the policy is represented in a neural network called the policy network and is assumed to be continuously differentiable with respect to its parameters $\theta$.

### 2.1    FRACTION OF VARIANCE EXPLAINED: $\mathcal{V}^{ex}$

The fraction of variance that the value function explains about the returns corresponds to the proportion of the variance in the dependent variable $V$ that is predictable from the independent variable $s_t$. We define $\mathcal{V}^{ex}_\tau$ as the *fraction of variance explained* for a trajectory $\tau$:

$$\mathcal{V}^{ex}_\tau \triangleq 1 - \frac{\sum_{t \in \tau} \left( \hat{R}_t - V(s_t) \right)^2}{\sum_{t \in \tau} \left( \hat{R}_t - \overline{R} \right)^2},  \tag{2}$$

where $\hat{R}_t$ and $V(s_t)$ are respectively the return and the expected return from state $s_t \in \tau$, and $\overline{R}$ is the mean of all returns in trajectory $\tau$. In statistics, this quantity is also known as the coefficient of determination $R^2$ and it should be noted that this criterion may be negative for non-linear models (Kvålseth, 1985), indicating a severe lack of fit of the corresponding function:

- $\mathcal{V}^{ex}_\tau = 1$: $V$ perfectly explains the returns - $V$ and the returns are *correlated*;
- $\mathcal{V}^{ex}_\tau = 0$ corresponds to a simple average prediction - $V$ and the returns are *not correlated*;
- $\mathcal{V}^{ex}_\tau < 0$: $V$ provides a worse fit to the outcomes than the mean of the returns.

One can have the intuition that $\mathcal{V}^{ex}_\tau$ close to 1 is interesting because it gives samples from an exercised behavior while $\mathcal{V}^{ex}_\tau < 0$ corresponds to a high mean-squared error for the value function, meaning the agent will learn from the corresponding samples. On the other hand, $\mathcal{V}^{ex}_\tau$ close to 0 does not provide any valuable information for a correct fitting of the state value function. We will demonstrate that $\mathcal{V}^{ex}$ is a relevant indicator for assessing self-performance in RL.

## 2.2 POLICY GRADIENT METHODS: PPO AND A2C

We use PPO (Schulman et al., 2017) throughout this work. We also use A2C, a synchronous variant of Mnih et al. (2016), to demonstrate SAUNA's performance. Below we provide some details regarding PPO, as we use it generously in this work. In previous work, PPO has been tested on a set of benchmark tasks and has produced impressive results in many cases despite a relatively simple implementation. At each iteration, the new policy $\theta_{new}$ is obtained from the old policy $\theta_{old}$:

$$\theta_{new} \leftarrow \underset{\theta}{\operatorname{argmax}} \underset{s_t, a_t \sim \pi_{\theta_{old}}}{\mathbb{E}} \left[ L^{\text{PPO}}\left(s_t, a_t, \theta_{old}, \theta\right) \right]. \tag{3}$$

We use the clipped version of PPO whose objective function is:

$$L^{\text{PPO}}(s_t, a_t, \theta_{old}, \theta) = \min\left( \frac{\pi_\theta(a_t|s_t)}{\pi_{\theta_{old}}(a_t|s_t)} A^{\pi_{\theta_{old}}}(s_t, a_t), \ g(\epsilon, A^{\pi_{\theta_{old}}}(s_t, a_t)) \right), \tag{4}$$

where

$$g(\epsilon, A) = \begin{cases} (1+\epsilon)A, A \geq 0 \\ (1-\epsilon)A, A < 0. \end{cases} \tag{5}$$

$A$ is the advantage function, $A(s, a) \triangleq Q(s, a) - V(s)$. The expected advantage function $A^{\pi_{\theta_{old}}}$ is estimated by an old policy and then re-calibrated using the probability ratio between the new and the old policy. In Eq. 4, this ratio is constrained to stay within a small interval around 1, making the training updates more stable.

## 2.3 RELATED WORK

Our method incorporates three key ideas: (a) function approximation with a neural network combining or separating the actor and the critic with an on-policy setting, (b) transition dropout reducing signal dilution in gradient updates while simplifying the underling MDP and (c) using $\mathcal{V}_\tau^{ex}$ as a measure of correlation between the value function and the returns to allow for better sampling and more efficient learning. Below, we consider previous work building on some of these approaches.

Actor-critic algorithms essentially use the value function to alternate between policy evaluation and policy improvement (Sutton & Barto, 1998; Barto et al., 1983). In order to update the actor, many methods adopt the on-policy formulation (Peters & Schaal, 2008; Mnih et al., 2016; Schulman et al., 2017). However, despite their important successes, these methods suffer from sample complexity.

In the literature, research has also been conducted in prioritization sampling. While Schaul et al. (2016) makes the learning from experience replay more efficient by using the TD error as a measure of these priorities in an off-policy setting, our method directly selects the samples on-policy. Schmidhuber (1991) is related to our method in that it calculates the expected improvement in prediction error, but with the objective to maximize the intrinsic reward through artificial curiosity. Instead, our method estimates the expected fraction of variance explained and filters out some of the samples to improve the learning efficiency.

Finally, motion control in physics-based environments is a long-standing and active research field. In particular, there are many prior work on continuous action spaces (Schulman et al., 2016; Levine & Abbeel, 2014; Lillicrap et al., 2016; Heess et al., 2015) that demonstrate how locomotion behavior and other skilled movements can emerge as the outcome of optimization problems.

## 3 SAUNA: A DYNAMIC TRANSITION DROPOUT METHOD

### 3.1 ESTIMATING $\mathcal{V}^{ex}$

While sampling the environment, SAUNA rejects samples where $V(s_t)$ is not correlated with the return from $s_t$. Therefore, $\mathcal{V}_\tau^{ex}$ must be estimated at each timestep so we define $\mathcal{V}_\theta^{ex}(s_t)$ as the prediction of $\mathcal{V}_\tau^{ex}$ under parameters $\theta$ at state $s_t \in \tau$. In addition, for shared parameters configurations, an error term on the value estimation is added to the PPO objective. The final objective becomes:

$$L(s_t, a_t, \theta_{old}, \theta) = \mathbb{E}\left[ L^{\text{PPO}}(s_t, a_t, \theta_{old}, \theta) - c_1\left(V_\theta(s_t) - \hat{R}_t\right)^2 - c_2\left(\mathcal{V}_\theta^{ex}(s_t) - \mathcal{V}_\tau^{ex}\right)^2 \right], \tag{6}$$

where $c_1$ and $c_2$ are respectively the coefficient for the squared-error loss of the value function and of the fraction of variance explained function. For cases where the network is not shared between the policy and the value function, $\mathcal{V}_\tau^{ex}$ is added to the value function network. Appendix A illustrates how $\mathcal{V}_\theta^{ex}(s_t)$ is embedded in the original architecture. The rest of the network is unchanged, making it very easy to use SAUNA without altering the complexity of existing policy gradient methods.

## 3.2 USING $\mathcal{V}^{ex}$ FOR DYNAMIC TEMPORAL ABSTRACTION

Mechanistically, SAUNA results in dropping out several transitions from a trajectory $\tau$, before each gradient update, and as a function of the starting state $s_t$ of those transitions. The mechanism is analogous to the method of dropout in deep learning but articulated here by a dropout in the state space of the MDP. To evaluate the theoretical implications of the transition dropout, our method can be formulated using the Options framework (Sutton et al., 1999; Precup, 2000) thoroughly applied to policy gradient methods in Smith et al. (2018). Previous work shows that by reasoning at several levels of abstraction, reinforcement learning agents are able to infer, learn, and plan more effectively. We detail below how SAUNA can be theoretically understood using the Options framework.

In this work, we take semi-Markov options that may depend on the entire preceding sequence. The agent is given access to a set of options, indexed by $\sigma$. Interestingly, the framework can be reduced as follows: all options share the same policy, $\pi_\sigma(a_t|s_t) = \pi_\theta(a_t|s_t)$, introduced in section 2, the same initiation set and the same termination condition which we parameterize as the function $\beta_\theta(s_t)$. $\beta_\theta(s_t)$ represents the probability of terminating an option and is defined as follows:

$$\beta_\theta(s_t) = \begin{cases} 0 & \text{if } b_\theta(s_0,\ldots,s_t) \geq r \\ 1 & \text{if } b_\theta(s_0,\ldots,s_t) < r \end{cases} \quad \text{with } b_\theta(s_0,\ldots,s_t) = \frac{|\mathcal{V}_t^{ex}|}{|\tilde{\mathcal{V}}_{0:t-1}^{ex}| + \epsilon_0}, \quad (7)$$

where for simplicity we rewrite $\mathcal{V}_\theta^{ex}(s_t)$ as $\mathcal{V}_t^{ex}$, $\tilde{\mathcal{V}}_{0:t-1}^{ex}$ is the median of $\mathcal{V}_\theta^{ex}$ between timesteps 0 and $t-1$, $\epsilon_0 = 10^{-8}$ is to avoid division by zero and $r$ is the dropout parameter. One may legitimately ask why not use directly $b_\theta(s_t) = |\mathcal{V}_t^{ex}|$. The rationale is practical: the ratio becomes a standardized measure as the agent learns, stabilized by the median, more robust to outliers than the mean.

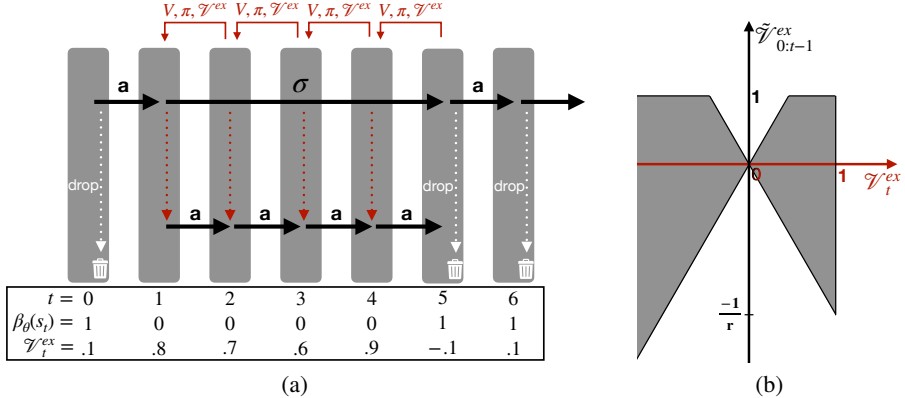

|  | $t = 0$ | 1 | 2 | 3 | 4 | 5 | 6 |
|---|---|---|---|---|---|---|---|
| $\beta_\theta(s_t) =$ | 1 | 0 | 0 | 0 | 0 | 1 | 1 |
| $\mathcal{V}_t^{ex} =$ | .1 | .8 | .7 | .6 | .9 | −.1 | .1 |

(a)             (b)

Figure 1: (a) Red solid arrows illustrate the transitions affecting the gradient update. We illustrate the termination condition mechanism with an example. For $t = (1, 2, 3, 4)$, $\mathcal{V}_t^{ex}$ is large enough so $\beta_\theta(s_t) = 0$. The transitions sampled by policy $\pi_\sigma$ affect the gradient update. The others are dropped. Note that an option $\sigma$ can be one-step long, i.e. $\beta_\theta(s_t) = 1, \beta_\theta(s_{t+1}) = 0, \beta_\theta(s_{t+2}) = 1$. (b) Grey area samples: kept for the gradient update. White area samples: dropped ($r = 0.3$).

Fig. 1a illustrates an example of transition dropout dynamics in SAUNA while Fig. 1b depicts the behavior of Eq. 7. The termination function $\beta_\theta(s_t)$ dynamically selects the transitions for which $\mathcal{V}_t^{ex}$ is either high or low, but not in between. Those transitions should have a strong impact on learning: a high absolute score means that the samples correspond to a state $s_t$ for which the value function is highly correlated with the returns or does not fit them properly. With this modification, the resulting gradients should be affected accordingly, which we investigate in section 4.2.2.

### 3.3 PUTTING IT ALL TOGETHER: SAUNA SAMPLING PROCEDURE

In this work, we hypothesize that low correlated transitions dilute the valuable signal information of the transitions with high absolute correlation. The temporal abstraction provided by the Options framework is leveled dynamically thanks to a termination function exploiting the self-performance measure $\mathcal{V}_t^{ex}$. Instead of keeping these noisy transitions, the sampling procedure drops them until the trajectory is $T$-steps long. Algorithm 1 illustrates how learning is achieved with SAUNA applied to PPO, in particular, the fitting of the $\mathcal{V}^{ex}$ function in Eq. 10 and the transition dropout in the *if* statement. We choose to depict a configuration where the parameters between the policy network, the value function and the $\mathcal{V}^{ex}$ function are not shared, since from this configuration the shared parameter case is direct.

---

**Algorithm 1** SAUNA sampling procedure in PPO.

---

**Initialize** policy parameters $\theta_0$
**Initialize** value function parameters $\phi_0$ and $\mathcal{V}^{ex}$ function parameters $\psi_0$

**for** $k = 0, 1, 2, \ldots$ **do**                                      $\triangleright$ For each update $k$

    **Initialize** trajectory $\tau$ to capacity T
    **while** $\text{size}(\tau) \leq T$ **do**                       $\triangleright$ For each timestep $t$
        $a_t \sim \pi_{\theta_k}(s_t)$, $v_t = V_{\phi_k}(s_t)$, $\mathcal{V}_t^{ex} = \mathcal{V}_{\psi_k}^{ex}(s_t)$
        **execute** action $a_t$ and observe reward $r_{t+1}$ and next state $s_{t+1}$
        **if** $\frac{|\mathcal{V}_t^{ex}|}{|\bar{\mathcal{V}}_{0:t-1}^{ex}|+\epsilon_0} \geq r$ **then**        PPO sampling
            **collect** transition $(s_t, a_t, r_t, v_t, s_{t+1}, \mathcal{V}_t^{ex})$ in $\tau$
        **else**
            **continue** without collecting the transition     SAUNA dropout

    **Gradient Update**

$$\theta_{k+1} \leftarrow \underset{\theta}{\arg\max} \sum_{t \in \tau} \min \left( \frac{\pi_\theta(a_t|s_t)}{\pi_{\theta_k}(a_t|s_t)} A^{\pi_{\theta_k}}(s_t, a_t), g(\epsilon, A^{\pi_{\theta_k}}(s_t, a_t)) \right) \tag{8}$$

$$\phi_{k+1} \leftarrow \underset{\phi}{\arg\min} \sum_{t \in \tau} \left( V_{\phi_k}(s_t) - \hat{R}_t \right)^2 \tag{9}$$

$$\psi_{k+1} \leftarrow \underset{\psi}{\arg\min} \sum_{t \in \tau} \left( \mathcal{V}_{\psi_k}^{ex}(s_t) - \hat{\mathcal{V}}_\tau^{ex} \right)^2 \tag{10}$$

---

## 4 HOW DOES SAUNA LEARN?

For ease of reproducibility and sharing, we have forked the original *baselines* repository from OpenAI and modified the code to incorporate our method[1]. The complete list of hyperparameters and details of our implementation are given in Appendix E and F respectively. A discussion about additional experiments whose results are non-positive, but which we think contribute positively to this paper, can be found in Appendix B.

### 4.1 SAUNA IN THE CONTINUOUS DOMAIN: MUJOCO AND ROBOSCHOOL

To verify the generalizability of our method, we study SAUNA against PPO and A2C. We compare SAUNA (PPO+$\mathcal{V}^{ex}$ in red) with its natural baseline PPO (PPO in blue). We use 6 simulated robotic tasks from OpenAI Gym (Brockman et al., 2016) using *MuJoCo* (Todorov et al., 2012). The two hyperparameters required by our method ($r = 0.3$ from Eq. 2 and $c_2 = 0.5$ from Eq. 6) and all the others (identical to those in Schulman et al. (2017)) are exactly the same for all tasks. We made this

---

[1]Code is available here: https://github.com/iclr2020-submission/denoising-gradient-updates

choice within a clear and objective framework of comparison between the two methods. Thus, we have not optimized the rest of the hyperparameters for SAUNA, and its reported performance is not necessarily the best that could be obtained with more intensive tuning.

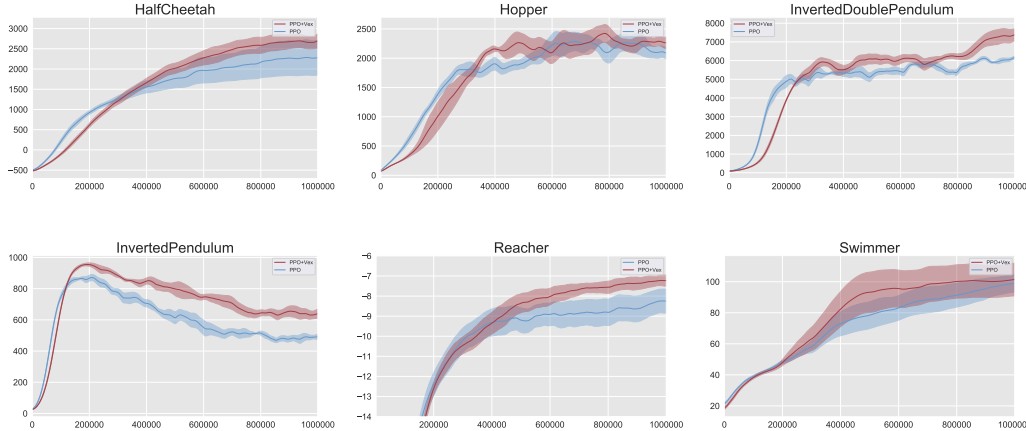

Figure 2: Comparison of SAUNA with PPO on 6 MuJoCo environments ($10^6$ timesteps, 6 different seeds). Red is our method PPO+$\mathcal{V}^{ex}$. Line: average performance. Shaded area: standard deviation.

From the graphs reported in Fig. 2, we see that our method outperforms all continuous control tasks. We also present in Table 1 the scores obtained for each task on the experiments with PPO and A2C. The graphs for A2C are reported in Appendix C.1.

Table 1: Average total reward of the last 100 episodes over 6 runs on the 6 MuJoCo environments on PPO and A2C. **Boldface** $mean \pm std$ indicate better mean performance.

| Task | PPO | PPO+$\mathcal{V}^{ex}$ | A2C | A2C+$\mathcal{V}^{ex}$ |
|---|---|---|---|---|
| HalfCheetah | $2277 \pm 432$ | $\mathbf{2929 \pm 169}$ | $1389 \pm 157$ | $\mathbf{1731 \pm 147}$ |
| Hopper | $2106 \pm 133$ | $\mathbf{2250 \pm 73}$ | $1367 \pm 110$ | $\mathbf{1627 \pm 97}$ |
| InvertedDoublePendulum | $6100 \pm 143$ | $\mathbf{6893 \pm 350}$ | $4151 \pm 67$ | $\mathbf{5132 \pm 409}$ |
| InvertedPendulum | $532 \pm 19$ | $\mathbf{609 \pm 24}$ | $686 \pm 15$ | $684 \pm 10$ |
| Reacher | $-7.5 \pm 0.8$ | $\mathbf{-7.2 \pm 0.3}$ | $-9.2 \pm 0.8$ | $\mathbf{-8.5 \pm 0.7}$ |
| Swimmer | $99.5 \pm 5.4$ | $100.8 \pm 10.4$ | $44.1 \pm 10.3$ | $\mathbf{59.0 \pm 5.5}$ |

We then experiment with the more difficult, high-dimensional continuous domain environment of *Roboschool* (Klimov & Schulman, 2017) with different neural network capacities for the model. When resources are limited in terms of number of parameters, it seems natural that dropping out samples based on their predicted learning impact allows to reduce noise in the gradient update and accelerate learning. And as a result, our method is faster and more efficient than the baseline. When resources are not limited, the gap closes towards the end of the training and our method performs as well as the baseline.

## 4.2 UNDERSTANDING THE IMPACT OF SAUNA ON LEARNING

### 4.2.1 WHAT ADVANTAGE DOES DROPOUT GIVE?

We further study the impact of dropping out noisy samples by conducting additional experiments in predicting $\mathcal{V}^{ex}$ while omitting the filtering step before the gradient update: the *if* statement in Algorithm 1 is removed and all transitions are collected in $\tau$. Indeed, the SAUNA algorithm could improve the agent's performance by simply training the shared network to optimize the variance explained head. Fig. 3 (full results are provided in Appendix D) demonstrates the positive effects of dropping out the samples. In addition, we studied the number of sample dropouts per task and their evolution throughout the training. On average, SAUNA rejects 5-10% of samples at the beginning of training which reduces to 2-6% at the end.

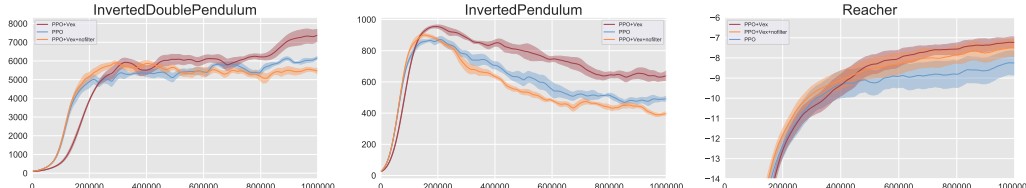

Figure 3: Comparison of SAUNA with PPO on 3 MuJoCo environments ($10^6$ timesteps, 6 different seeds). Red is our method PPO+$\mathcal{V}^{ex}$, Orange is PPO+$\mathcal{V}^{ex}$ without the filtering out of noisy samples. Line: average performance. Shaded area: standard deviation.

### 4.2.2 HOW DOES SAUNA IMPACT THE GRADIENTS?

Prior to the gradient updates, SAUNA removes the transitions for which the state value is poorly correlated with the returns. By doing so, we hypothesized that information signals from samples with significant $\mathcal{V}^{ex}$ would be less diluted by dropped out samples. One can easily see by the graphs that the norm of the gradients is affected by such a change in the learning procedure. Fig. 4 shows that SAUNA dropout generates larger gradients. Hence policy updates have bigger steps, which eventually results in better performance.

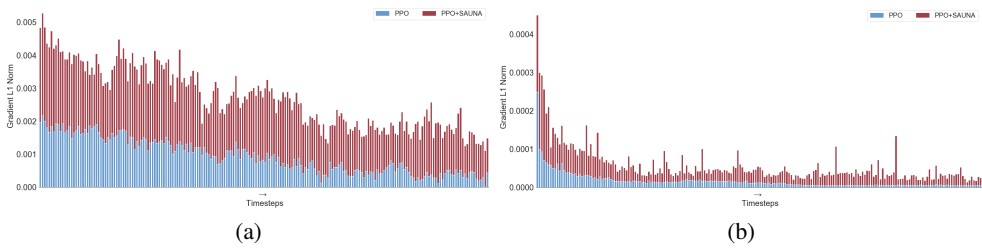

Figure 4: Gradients L1-norm from the (a) first layer and (b) last layer of the shared parameters network for PPO and when SAUNA is applied to PPO. Task: HalfCheetah-v2.

One may wonder quite rightly why performance is not damaged, since larger gradients could hinder learning. The performance results suggest that the gradients contain more useful information from each of the transitions that passed SAUNA sampling. In other terms, the relevant information for the task at hand is less diluted, the gradients are more qualitative and have been partially denoised.

### 4.2.3 HALFCHEETAH: A QUALITATIVE STUDY

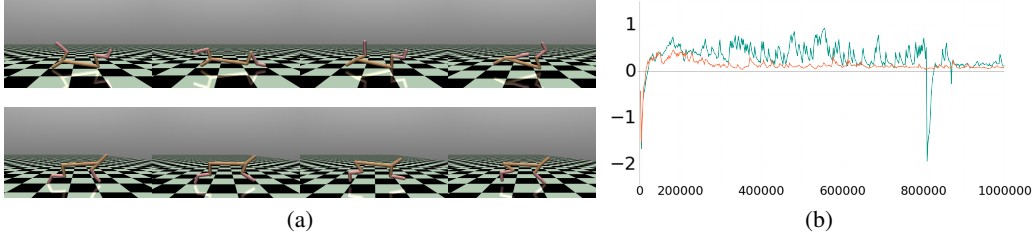

Figure 5: (a) Example of PPO getting trapped in a local minimum (top row) while PPO+$\mathcal{V}^{ex}$ reaches a better optimum (bottom row). (b) $\mathcal{V}^{ex}$ score for PPO (orange) and SAUNA (green).

In *HalfCheetah*, a well-known behavior (Lapan, 2018) is that for multiple seeds PPO is converging to a local minimum forcing the agent to move forward on its back. However, we observed that SAUNA made it possible to leave from, or at least to avoid these local minima. This is illustrated in Fig. 5a where we can see still frames of two agents trained with PPO and SAUNA for $10^6$ timesteps on

identically seeded environments. Their behavior is entirely different. Looking at $\mathcal{V}^{ex}$ in Fig. 5b, we can see that the graphs differ quite interestingly. The orange agent seems to find very quickly a local minimum on its back while the green agent's $\mathcal{V}^{ex}$ varies much more. This seems to allow the latter to explore more states than the former and finally find the fastest way forward. Supported by the previous study, we can infer that SAUNA agents are better able to explore interesting states while exploiting with confidence the value given to the states observed so far.

## 5 DISCUSSION

Intuitively, for the policy update, our method will only use qualitative samples that provide the agent with (a) reliable and exercised behavior (high $\mathcal{V}^{ex}$) and (b) challenging states from the point of view of correctly predicting their value (low $\mathcal{V}^{ex}$). The SAUNA algorithm keeps samples with high learning impact, rejecting other noisy samples from the gradient update.

### 5.1 DENOISING POLICY GRADIENT UPDATES AND THE POLICY GRADIENT THEOREM

Policy gradient algorithms are backed by the policy gradient theorem (Sutton et al., 2000):

$$\nabla_\theta L(\theta) = \propto \sum_{s \in S} d^\pi(s) \sum_{a \in \mathcal{A}} Q^\pi(s,a) \nabla_\theta \pi_\theta(a|s). \tag{11}$$

As long as the asymptotic stationary regime is not reached, it is not reasonable to assume the sampled states to be independent and identically distributed (i.i.d.). Hence, it seems intuitively better to ignore some of the samples for a certain period, to allow the most efficient use of information. One can understand SAUNA as making gradient updates more robust through dropout, especially when the update is low and the noise can be dominant. Besides, not taking all samples reduces the bias in the state distribution $d^\pi$. Therefore, it now seems more reasonable to consider the sampled states i.i.d., which we theoretically need for the policy gradient theorem.

### 5.2 IMPACT OF $\mathcal{V}^{ex}$ ON THE SHARED NETWORK PARAMETERS

The shared network predicts $\mathcal{V}^{ex}$ in conjunction with the value function and the policy. Therefore, as its parameters are updated through gradient descent, they converge to one of the objective function minima (hopefully, a global minimum). This parameter configuration integrates $\mathcal{V}^{ex}$, predicting how much the value function has fit the observed samples, or informally speaking how well the value function is doing for state $s_t$. This new head tends to lead the network to adjust predicting a quantity relevant for the task. Instead of using domain knowledge for the task, the method rather introduces problem knowledge by constraining the parameters directly.

## 6 CONCLUSION

Policy gradient methods optimize the policy directly through gradient ascent with the objective to maximize the expected return. We have introduced a new, lightweight and agnostic method technically applicable to any policy gradient method using a neural network as function approximation. The central idea of this paper is that samples that are uncorrelated with the return are dropped out and not considered for the policy update. Those low correlated samples are ignored by SAUNA, the mechanism is reflected in a dynamic temporal abstraction controlled by the estimated variance explained of each state. The relevant signal information being less diluted, this results in a denoising effect on the gradients, ultimately leading to improved performance.

We demonstrated the effectiveness of our method when applied to two policy gradient methods on several standard benchmark environments. We also established that samples can be removed from the gradient update without hindering learning but can, on the opposite, improve it. We further studied the impact that such a modification in the sampling procedure has on learning. Several open topics warrant further study. The influence of SAUNA on the distribution of states could, for instance, be theoretically investigated. We also find the effort (Cobbe et al., 2019; Zhang et al., 2018) to go further towards generalization in RL very promising, and we think SAUNA could be useful in these problems as a way to regularize policy gradient methods.

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

# A ILLUSTRATION OF THE SAUNA ARCHITECTURE

Figure 6: Network-agnostic variance explained head.

In Fig. 6 we rewrite $\mathcal{V}^{ex}_\theta(s_t)$ as $\mathcal{V}^{ex}_t$. On the right side of the figure is illustrated the $\mathcal{V}^{ex}_t$ head that is added to the shared or the separate network configurations. Note that even if $\mathcal{V}^{ex}_\tau$ is defined for a sampled trajectory $\tau$, the model predicts its value at each state $s_t \in \tau$.

# B ADDITIONAL EXPERIMENTS WITH NON-POSITIVE RESULTS

**Atari domain.** We tested our method on the *Atari 2600* domain (Bellemare et al., 2013) without observing any improvement in learning. By comparing the two algorithms where the method of filtering the samples is used or not, we could not observe any difference, as some tasks were better performed by one method and others by the other.

**Mean of $\mathcal{V}^{ex}$.** Although $\tilde{\mathcal{V}}^{ex}$, the median of $\mathcal{V}^{ex}$, is more expensive to calculate, we observe that it gives much better results than if we use its mean in Eq. 7. Using the median helps (Kvålseth, 1985) because the distribution of $\mathcal{V}^{ex}$ is not normal and includes outliers that will potentially produce misleading results.

**Non-empirical $\mathcal{V}^{ex}$.** We also experimented with using the real values of $\mathcal{V}^{ex}$ in Eq. 7 when calculating $\tilde{\mathcal{V}}^{ex}_{0:t-1}$, instead of the predicted ones. This has yielded less positive results, and it is likely that this is due to the difference between the predicted and actual values at the beginning of learning, which has the effect of distorting the ratio in Eq. 7.

**Adjusting state count.** In order to stay in line with the policy gradient theorem (Sutton et al., 2000), we have worked to adjust the distribution of states $d^\pi$ to what it really is, since some states that the agent has visited are not included in the gradient update. We adjusted it using the ratio between the number of states visited and the actual number of transitions used in the gradient update, but this did not improve the learning, and instead, we observed a decrease in performance.

**Random dropout.** We experimented with dropping out at random, and before each gradient update, a number of samples corresponding to the same approximate number of samples that SAUNA drops. This resulted in a decrease in performance compared to PPO, as one can expect.

# C  ADDITIONAL EXPERIMENTS WITH SAUNA IN THE CONTINUOUS DOMAIN

## C.1  MUJOCO: COMPARISON OF A2C+SAUNA AND A2C

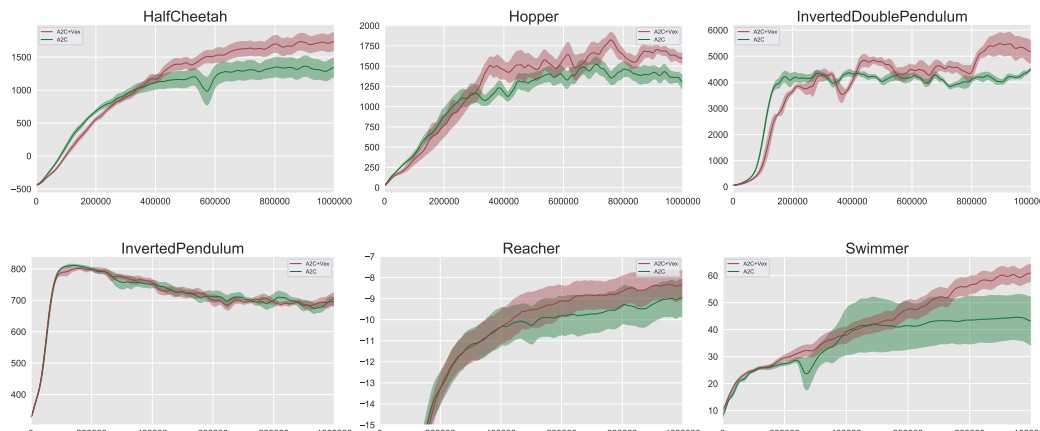

Figure 7: Comparison of SAUNA with A2C on 6 MuJoCo environments ($10^6$ timesteps, 6 different seeds). Red is our method A2C+$\mathcal{V}^{ex}$. Line: average performance. Shaded area: standard deviation.

In Fig. 7 we compare A2C+SAUNA (A2C+$\mathcal{V}^{ex}$ in red) with A2C (A2C in green) on again 6 simulated robotic tasks from OpenAI Gym using the MuJoCo physics engine.

## C.2  ROBOSCHOOL: COMPARISON OF PPO+$\mathcal{V}^{ex}$ AND PPO

We experiment here with the more difficult, high-dimensional continuous domain environment of *Roboschool* (Klimov & Schulman, 2017): *RoboschoolHumanoidFlagrunHarder-v1*. The purpose of this task is to allow the agent to run towards a flag whose position varies randomly over time. It is continuously bombarded by white cubes that pushes it out of its path, and if it does not hold itself up it is left to fall.

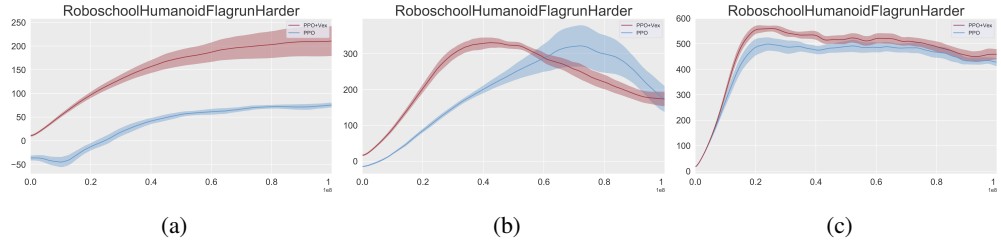

Figure 8: Comparison of SAUNA with PPO on the more challenging Roboschool environment ($10^8$ timesteps, 6 different seeds). Red is our method PPO+$\mathcal{V}^{ex}$. Line: average performance. Shaded area: standard deviation.

In Fig. 8a, the same fully-connected network as for the MuJoCo experiments (2 hidden layers each with 64 neurons) is used. In Fig. 8b, the network is composed of a larger 3 hidden layers with 512, 256 and 128 neurons. We trained those agents with 32 parallel actors. In both experiments, SAUNA performs better and learns faster at the beginning. Then, only when the policy and value functions benefit from a larger network, the gap closes, and our method does as well as the baseline. When resources are limited in terms of the number of parameters, it seems natural that filtering out samples based on their predicted training impact helps to reduce noise in the gradient update and accelerate learning.

Finally, we investigated further and conducted the same experiment with the larger network (3 hidden layers with 512, 256 and 128 neurons), but with 128 actors in parallel instead of 32. Results are reported in Fig. 8c: our method still learns faster and achieves better performance than the baseline.

## D  ADDITIONAL EXPERIMENTS ON THE ADVANTAGE OF TRANSITION DROPOUT

In order to identify the effects of the training of the $\mathcal{V}^{ex}$ head and the filtering out of sample, we verify the hypothesis that filtering out noisy samples does improves the performance. To do so, in Section 4.2.1, we conduct experiments where the network predicts $\mathcal{V}^{ex}$ but where the noisy samples are not filtered out: the *if* statement in Algorithm 1 is removed and all transitions are collected in $\tau$. In Fig. 9, we see that when the noisy samples are not filtered out the performance is worst than the baseline, confirming the positive denoising impact of filtering out the variance-selected samples.

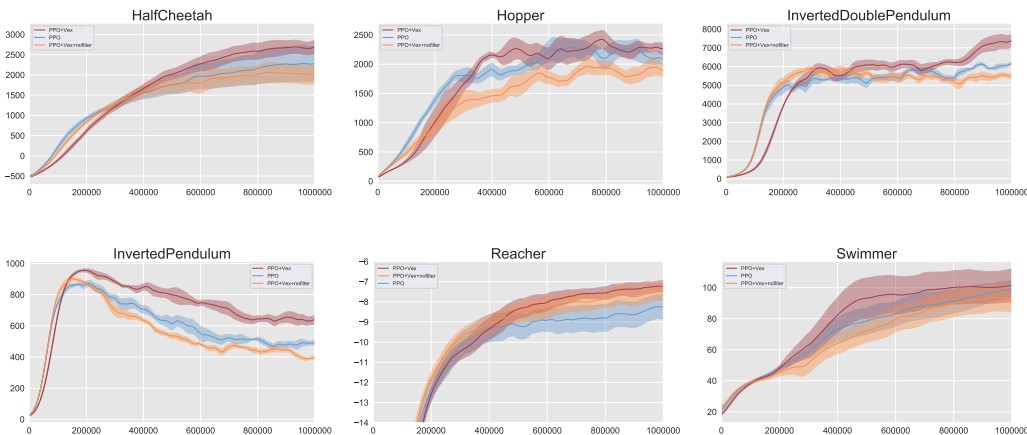

Figure 9: Comparison of our method with PPO on 6 MuJoCo environments ($10^6$ timesteps, 6 different seeds). Red is our method PPO+$\mathcal{V}^{ex}$, Orange is PPO+$\mathcal{V}^{ex}$ without the filtering out of noisy samples. Line: average performance. Shaded area: standard deviation.

## E  HYPERPARAMETERS

We have tuned the hyperparameters of our method by performing a grid search and selecting the best combinations by considering those with the largest consensus.

| Hyperparameter | Value |
|---|---|
| Horizon ($T$) | 2048 (MuJoCo), 512 (Roboschool) |
| Adam stepsize | $3 \cdot 10^{-4}$ |
| Nb. epochs | 10 (MuJoCo), 15 (Roboschool) |
| Minibatch size | 64 (MuJoCo), 4096 (Roboschool) |
| Discount ($\gamma$) | 0.99 |
| GAE parameter ($\lambda$) | 0.95 |
| Clipping parameter ($\epsilon$) | 0.2 |
| VF coef ($c_1$) | 0.5 |
| $\mathcal{V}^{ex}$ coef ($c_2$) | 0.5 |
| Dropout threshold ($r$) | 0.3 |

Table 2: Hyperparameters used both in PPO and SAUNA. The two last hyperparameters are only relevant for our method.

| Hyperparameter | Value |
|---|---|
| Horizon ($T$) | 2048 |
| Adam stepsize | $3 \cdot 10^{-4}$ |
| Nb. epochs | 10 |
| Minibatch size | 64 |
| Nb. envs | 8 |
| Discount ($\gamma$) | 0.99 |
| VF coef ($c_1$) | 0.5 |
| $\mathcal{V}^{ex}$ coef ($c_2$) | 0.5 |
| Dropout threshold ($r$) | 0.35 |

Table 3: Hyperparameters used both in A2C and SAUNA. The two last hyperparameters are only relevant for our method.

## F  IMPLEMENTATION DETAILS

Unless otherwise stated, the policy network used for MuJoCo and Roboschool tasks is a fully-connected multi-layer perceptron with 2 hidden layers of 64 units. For Atari, the network is shared between the policy and the value function and is the same as in Mnih et al. (2016). The architecture for the $\mathcal{V}^{ex}$ function head is the same as for the value function head.

## G  CLIPPED SURROGATE OBJECTIVE DETAILS

In Eq. 4, we use the following standard definitions for the advantage function:

$$A^{\pi}(s, a) = Q^{\pi}(s, a) - V^{\pi}(s), \tag{12}$$

where

$$Q^{\pi}(s_t, a_t) = \mathop{\mathbb{E}}_{\substack{s_{t+1:\infty} \\ a_{t+1:\infty}}}\left[\sum_{l=0}^{\infty} \gamma^l r_{t+l}\right] \text{ and } V^{\pi}(s_t) = \mathop{\mathbb{E}}_{\substack{a_{t:\infty} \\ s_{t+1:\infty}}}\left[\sum_{l=0}^{\infty} \gamma^l r_{t+l}\right]. \tag{13}$$

## H  AN ANALOGY WITH SAUNAS

Saunas originated in Northern Europe and are thought to date back to 7000 BC. Their use helps to release impurities [filtered out noisy samples] and improves the regeneration of cells [improved policy gradient updates]. Their temperatures could be fatal if not regulated by humidity [$\mathcal{V}^{ex}$ criterion].

