# OpenReview forum: "Samples Are Useful? Not Always: denoising policy gradient updates using variance explained"
_ICLR.cc/2020/Conference — Reject_

### Official Review · AnonReviewer1 · 2019-10-22
**Official Blind Review #1**

**Rating:** 6

**Review:**

This paper proposes a simple modification to policy gradient methods that relies on the variance explained to filter samples. The paper contains experiments showing empirical gains to the method, as well as some evidence that filtering rather than simply predicting more quantities is making a difference.

I put a "weak accept" score. This method is novel, afaict, and is based on interesting statistical hypotheses. I think showing why this method is relevant could be better executed. There are some gains compared to a PPO baseline, but the gains are somewhat incremental and may only apply to PPO, as other PG methods aren't tested.

Detailed comments:
- is this method always relevant? Are there environments where it makes more sense to use? (in terms of reward density/variance, exploration difficulty, etc.) Policy gradient algorithms for which it makes more sense to add?
- The conclusion in particular claims much more than what is in the paper:
>> "applicable to any policy gradient method", technically yes, but it remains untested
>> "SAUNA removes noise", also remains to be seen. A graph showing this would add a lot to this paper
>> "We [..] studied the impact [..] on both the exploitation [..] and on the exploration", Figure 5 is a single data point, where one run for one environment got out of a "well-known" local minima. To really convince readers of this you would need to test your method on multiple environments with actual exploration and sparse rewards (Mujoco locomotion tasks do not satisfy those requirements, even though they has local minima, this is far from exploration as commonly understood in RL)
- This method is very much related to prioritized experience replay and others, as observed in 2.2, yet no comparison is made (PER can be implemented online using two exponential moving averages to estimate the distribution of TD error, used to do rejection sampling). Simpler baselines could also have been tested against, e.g. simply rejecting samples with high TD error.
- It's not clear how the threshold of 0.3 was chosen, nor what its effect is empirically.
- Estimating V^ex vs using empirical V^ex seems to make a big difference. It's not obvious why we should be estimating V^ex at all, and I think this deserves more analysis.
- I've seen Roboschool cited, you can use @misc{klimov2017roboschool, title={Roboschool}, author={Klimov, Oleg and Schulman, J}, year={2017} }

**Experience Assessment:**

I have published one or two papers in this area.

**Review Assessment: Checking Correctness Of Derivations And Theory:**

I assessed the sensibility of the derivations and theory.

**Review Assessment: Checking Correctness Of Experiments:**

I carefully checked the experiments.

**Review Assessment: Thoroughness In Paper Reading:**

I read the paper at least twice and used my best judgement in assessing the paper.

---

> ### Author Response · Authors · 2019-11-15
> **Response to official review**
>
> Thank you for the thorough review. We have updated the paper based on your suggestions.
>
>
> > I think showing why this method is relevant could be better executed. There are some gains
> > compared to a PPO baseline, but the gains are somewhat incremental and may only apply to
> > PPO, as other PG methods aren't tested.
>
> Thank you for the great suggestions. We have restructured and reformulated both the motivation in Section 1 and Section 3, where we now clearly articulate the building blocks of the method. We have also, based on your comments, did additional experiments: we have included a comparison with another policy gradient method in Section 4.1 as well as an experiment in Section 4.2.2, exposing the gradients norm when using PPO or using SAUNA with PPO.
>
>
> We have addressed your detailed comments below and incorporated the discussion into the revised article.
>
> > Is this method always relevant? Are there environments where it makes more sense to use?
> > (in terms of reward density/variance, exploration difficulty, etc.) Policy gradient algorithms
> > for which it makes more sense to add?
>
> In order to strengthen the study of the relevance of our method, we have included a comparison with A2C in the revised version of the manuscript. Our experiments with Roboschool suggest that when the task is difficult and the model resources are low for the agent to learn, rejecting some samples based on whether they are correlated with the returns can help. PG algorithms with trust regions (such as TRPO or PPO) could, in practice, benefit more from a sample dropout than e.g., A2C. One of the reasons we think of is that, intuitively, algorithms constraining the new policy to stay within a small interval from the old one need richer and more qualitative gradients in order to shift the agent parameters with an amplitude similar to other methods. Although, in practice, we did not observe such results that support this.
>
>
> > "applicable to any policy gradient method", technically yes, but it remains untested
>
> Indeed, we meant “applicable” in the technical sense here, sorry for the confusion. We clarified this in the revised manuscript. In response to this very legitimate comment, we tested the method on an additional policy gradient method, which we think strengthens the paper. The experiments are included in Section 4.1 and Appendix B.1.
>
>
> > "SAUNA removes noise", also remains to be seen. A graph showing this would
> > add a lot to this paper
>
> Thank you for the great comment, we have reformulated the claim. In addition, we have clarified the implications of using V^ex as a way to reject samples uncorrelated with the returns in Section 2.1, and included a justification for this claim by looking at the modifications SAUNA brings to the gradients in Section 4.2.2.
>
>
> > "We [..] studied the impact [..] on both the exploitation [..] and on the exploration”,
> > Figure 5 is a single data point, where one run for one environment got out of a "well-known"
> > local minima
>
> Thank you for pointing out this. We have removed this claim in the conclusion of the revised manuscript and clarified the interpretation of the qualitative study in Section 4.2.3.
>
>
> > This method is very much related to prioritized experience replay and others, as observed in 2.2,
> > yet no comparison is made (PER can be implemented online using two exponential moving averages
> > to estimate the distribution of TD error, used to do rejection sampling). Simpler baselines could also
> > have been tested against, e.g. simply rejecting samples with high TD error.
>
> Thank you for your suggestion. Because of time constraints, we were required instead to do more experiments on an additional policy gradient method, as the suggestion was unanimous among the Reviewers, and this would improve a lot the paper. Although, for the sake of thoroughness, we also experimented with randomly dropping out samples, and we include the results in Appendix D.
>
>
> > It's not clear how the threshold of 0.3 was chosen, nor what its effect is empirically.
>
> We have tuned the hyperparameters of our method by performing a grid search and selecting the best combinations by considering those with the largest consensus. We have included this information in the manuscript in Appendix E. We have clarified the effects of thresholding in Section 3.2.
>
>
> > Estimating V^ex vs using empirical V^ex seems to make a big difference. It's not obvious
> > why we should be estimating V^ex at all, and I think this deserves more analysis.
>
> Thank you for the suggestion. Vex should be fit by a parametric function because we need to estimate the value of Vex(s_{t}) for each state s_{t} belonging to a trajectory \tau. Indeed, V^ex is defined for a trajectory. We clarify this in Section 3.1.
>
>
> > I've seen Roboschool cited, you can use @misc{klimov2017roboschool, title={Roboschool},
> > author={Klimov, Oleg and Schulman, J}, year={2017} }
>
> Thank you, we have corrected this in revised manuscript.

---

### Official Review · AnonReviewer3 · 2019-10-23
**Official Blind Review #3**

**Rating:** 3

**Review:**

This paper proposes a novel way to denoise the policy gradient by filtering the samples to add by a criterion "variance explained". The variance explained basically measures how well the learn value function could predict the average return, and the filter will keep samples with a high or low variance explained and drop the middle samples. This new mechanism is then added on top of PPO to get their algorithm SAUNA. Empirical results show that it is better than PPO, on a set of MuJoCo tasks and Roboschool.

From my understanding, this paper does not show a significant contribution to the related research area. The main reason I tend to reject this paper is that the motivation of their proposed algorithm is very unclear, lack of theoretical justification and the empirical justification is restricted on PPO -- one policy gradient method.

1) It's unclear to me how it goes to the final algorithm, and what is the intuition behind it. Second 3.1 is easy to follow but the following part seems less motivated. In section 3.2 it's unclear to me why we need to fit a parametric function of Vex. In section 3.2, it's unclear to me why the filter condition is defined as Eq (7). The interpretation is a superficial explanation of what Eq 7 means but does not explain why I should throw out some of my samples, why high and low score means samples are helpful for learning and score in between does not?

2) This paper argues the filter condition improves PG algorithms by denoising the policy gradient. This argument is not justified at all except a gradient noise plot in one experimental domain in figure 5b. That's not enough to support the argument that what this process is really doing. Some theoretical understanding of what the dropped/left samples will do is helpful.

3) The method of denoising the policy gradient is expected to help policy gradient methods in general. It's important to show at least one more PG algorithm (DDPG/REINFORCE/A2C) where the proposed method can help, for verifying the generalizability of algorithm.

In general, I feel that the content after section 3.1 could be presented in a much more principled way. It should provide not just an algorithm and some numbers in experiments, but also why we need the algorithm, what's the key insight of designing this algorithm, what the algorithm really did by the algorithm mechanism itself instead of just empirical numbers.


**Experience Assessment:**

I have published one or two papers in this area.

**Review Assessment: Checking Correctness Of Derivations And Theory:**

N/A

**Review Assessment: Checking Correctness Of Experiments:**

I assessed the sensibility of the experiments.

**Review Assessment: Thoroughness In Paper Reading:**

I read the paper at least twice and used my best judgement in assessing the paper.

---

> ### Author Response · Authors · 2019-11-15
> **Response to official review**
>
> Thank you for the thoughtful review. These suggestions and questions are reflected in the updated article.
>
> > The main reason I tend to reject this paper is that the motivation of their
> > proposed algorithm is very unclear, lack of theoretical justification and
> > the empirical justification is restricted on PPO -- one policy gradient method.
>
> We have restructured Section 2 and Section 3 in order to clarify the motivation for the method, and we now clearly articulate the building blocks of the method as well as the theoretical grounding for the transition dropout. We have enriched the paper with additional experiments in Section 4.1 - with experiments on an additional policy gradient method - and Section 4.2 - with a study of the impact of our method on the gradients.
>
>
> We have addressed your key point below and incorporated the discussion into the revised article.
>
> 1. Thank you for your great comments and question. We have restructured Section 3. In Section 3.1, we clarify that Vex should be fit by a parametric function because we need to estimate the value of V^ex(s_{t}) for each state s_{t} belonging to a trajectory \tau. We also clarify the rationale behind Eq. 7. We also give in Section 2.1 a more thorough justification for the rejection of samples for which the state value function and the returns are not correlated.
>
> 2. Thank you for this important observation. In order to study more thoroughly the implication of such a change in the sampling algorithm, we have added in Section 4.2.2 an experiment exposing the L1-norm of the gradients for PPO and when SAUNA is applied to PPO. Those graphs, together with the performance results, suggest that the gradients contain more useful information from each of the transitions that passed SAUNA sampling. This also infers that the temporal abstraction introduced in the policy gradient enriches the gradients with more qualitative information and that they have been partially denoised.
>
> 3. We have compared our method with an additional policy gradient algorithm, A2C, and we report the (positive) results in Section 4.1 and Appendix B.1.
>
>
> > In general, I feel that the content after section 3.1 could be presented in a much more principled way.
>
> Based on your suggestion, in addition to restructuring Section 3, we have enriched and presented Section 4 in a more diligent way.

---

### Official Review · AnonReviewer2 · 2019-10-24
**Official Blind Review #2**

**Rating:** 6

**Review:**

In their post-review revision, the authors have added a much clearer motivation for SAUNA, along with extra experiments that validate and clarify the approach.

The revision is a significantly better paper than the original, and I am updating my score accordingly.

-------------

This paper introduces 'the fraction of variance explained' V^{ex} as a measure of the ability of the value function to fit the true reward. Given the mean per-timestep reward as a baseline, V^{ex} measures the proportion of the variance in reward that is captured by the value function.

In this paper, the authors introduce a filtering condition aimed to ensure that no single transition excessively reduces the fraction of variance explained. This filtering condition relies on a prediction of the variance explained, which comes from an extra head added to the standard PPO architectures, and its parameters are updated along with all other model parameters at the end of each trajectory.

Intuitively, I can believe that learning will be more stable under the condition that no single transition leads to too great a divergence between the predicted reward and true reward. However, I do not understand the authors' assertion that this filtering procedure removes noisy samples (how can we characterize these samples as noise?). I'm also insufficiently familiar with the related literature to properly gauge the theoretical implications of this modification (see low confidence below).

The paper compares learning accuracy over time of PPO with and without the variance explained filtering. It seems that the filtering does improve learning for the MuJoCo environments, as well as low resource models for a harder task from the Roboschool environment but it is hard to state definitively that the new method is better. I commend the authors on their discussion of non-positive Atari results in the appendix and I agree that it contributes significantly to the paper.

With the caveat that this paper is quite far from the realm of my expertise. I think that the approach is intriguing but, in the absence of any theoretical justification for the approach, I'm not sure that the empirical results are sufficiently convincing for ICLR. I also believe that the paper would be easier to understand with a more thorough investigation of the effect of the filtering on the learning procedure.


Questions for the authors:

- what proportion of samples are rejected

- how does this proportion change over the course of learning

- how was the V^{ex} threshold of 0.3 chosen

**Experience Assessment:**

I do not know much about this area.

**Review Assessment: Checking Correctness Of Derivations And Theory:**

I assessed the sensibility of the derivations and theory.

**Review Assessment: Checking Correctness Of Experiments:**

I assessed the sensibility of the experiments.

**Review Assessment: Thoroughness In Paper Reading:**

I read the paper at least twice and used my best judgement in assessing the paper.

---

> ### Author Response · Authors · 2019-11-15
> **Response to official review**
>
> Thank you for the clear and encouraging review. We have addressed your key points below and incorporated the discussion into the revised article.
>
>
> > However, I do not understand the authors' assertion that this filtering procedure
> > removes noisy samples (how can we characterize these samples as noise?).
>
> Thank you for the great comment (comment shared with Reviewer 1 and Reviewer 3). We have updated the manuscript to support this claim and included in Section 4.2.2 an additional experiment exposing the L1-norm of the gradients throughout the learning for PPO and for SAUNA applied to PPO. The important difference in norm suggests that the gradients are enriched with more informative samples allowing the algorithm to do bigger steps towards a better policy.
>
>
> > The paper compares learning accuracy over time of PPO with and without the variance
> > explained filtering. It seems that the filtering does improve learning for the MuJoCo environments,
> > as well as low resource models for a harder task from the Roboschool environment but it is hard to
> > state definitively that the new method is better.
>
> Thank you for this observation. In order to verify the generalizability of our method and strengthen the paper, we have added a comparison with another policy gradient method, A2C. The results have been added in Section 4.1. We have also clarified and better emphasized the positive implications of such a change in the sampling procedure in Section 3 and 4.
>
>
> > I think that the approach is intriguing but, in the absence of any theoretical justification for the approach,
> > I'm not sure that the empirical results are sufficiently convincing for ICLR. I also believe that the paper would
> > be easier to understand with a more thorough investigation of the effect of the filtering on the learning procedure.
>
> We have reformulated the theoretical grounding of the transition dropout introduced by our method in Section 3.2, which allows for reframing the sample dropout as a dynamic temporal abstraction, which we know from [1] that one way to improve on existing agents is to leverage abstraction. We have also included in Section 4.2.2 an additional experiment we referred to in the first point.
>
> [1] Smith, Hoof, and Pineau. An inference-based policy gradient method for learning options. ICML. 2018.
>
>
> Thank you for the detailed questions. The answers are reflected in the updated article.
>
> > What proportion of samples are rejected and how does this proportion
> > change over the course of learning?
>
> On average, SAUNA rejects 5-10% of samples at the beginning of training which reduces to 2-6% at the end. We have included this information in the manuscript in Section 4.2.1.
>
> > How was the V^{ex} threshold of 0.3 chosen?
>
> We have tuned the hyperparameters of our method by performing a grid search and selecting the best combinations by considering those with the largest consensus. We have also included this information in the manuscript in Appendix E.

---

### Author Response · Authors · 2019-11-15
**General response to official reviews**

We thank the reviewers for their time and thoughtful feedback.

We have updated the submission. We have clarified the motivation in the introduction and restructured section 3, where the method is now more carefully unveiled. We think that this will help to highlight better the relevance of the method. We have clarified the rationale for rejecting some of the samples to accelerate learning, added experiments on another policy gradient method to strengthen the paper, included a more theoretical investigation with temporal abstraction on the effect of sample dropout during the learning procedure, and we have resolved ambiguities regarding the denoising impact of SAUNA on the gradients.

We answer specific questions raised in the reviews by separately replying to each of them.

---

### Decision · Program_Chairs · 2019-12-19

**Decision:**

Reject

**Comment:**

The authors aim to improve policy gradient methods by denoising the gradient estimate. They propose to filter the transitions used to form the gradient update based on a variance explains criterion. They evaluate their method in combination with PPO and A2C, and demonstrate improvements over the baseline methods.

Initially, reviewers were concerned about the motivation and explanation of the method. The authors revised the paper by clarifying the motivation and providing a justification based on the options framework. Furthermore, the authors included additional experiments investigating the impact of their approach on the gradient estimator, showing that with their filtering, the gradient estimator had larger magnitude.

Reviewers found the justification via the options framework to be a stretch, and I agree. The authors should explain how the options framework leads to dropping gradient terms. At the moment, the paper describes an algorithm using the options framework, however, they don't connect the policy gradients of that algorithm to their method. Furthermore, the authors should more clearly verify the claims about reducing noise in the gradient estimate. While the additional experiments on the norm are nice, the authors should go further. For example, if the claim is that the variance of the gradient estimator is reduced, then that should be verified. Finally, there are many approaches for reducing the variance of the policy gradient (Grathwohl et al. 2018, Wu et al 2018, Liu et al. 2018) and no comparisons are made to these approaches.

Given the remaining issues, I recommend rejection for this paper at this time, however, I encourage the authors to address these issues and submit to a future venue.